# Learning From Brains How to Regularize Machines

**Zhe Li[1-2,\*], Wieland Brendel[4-5], Edgar Y. Walker[1-2,7], Erick Cobos[1-2],**
**Taliah Muhammad[1-2], Jacob Reimer[1-2], Matthias Bethge[4-6],**
**Fabian H. Sinz[2,5,7], Xaq Pitkow[1-3], Andreas S. Tolias[1-3]**

[1] Department of Neuroscience, Baylor College of Medicine
[2] Center for Neuroscience and Artificial Intelligence, Baylor College of Medicine
[3] Department of Electrical and Computer Engineering, Rice University
[4] Centre for Integrative Neuroscience, University of Tübingen
[5] Bernstein Center for Computational Neuroscience, University of Tübingen
[6] Institute for Theoretical Physics, University of Tübingen
[7] Institute Bioinformatics and Medical Informatics, University of Tübingen

[\*]zhel@bcm.edu

## Abstract

Despite impressive performance on numerous visual tasks, Convolutional Neural Networks (CNNs) — unlike brains — are often highly sensitive to small perturbations of their input, *e.g.* adversarial noise leading to erroneous decisions. We propose to regularize CNNs using large-scale neuroscience data to learn more robust neural features in terms of representational similarity. We presented natural images to mice and measured the responses of thousands of neurons from cortical visual areas. Next, we denoised the notoriously variable neural activity using strong predictive models trained on this large corpus of responses from the mouse visual system, and calculated the representational similarity for millions of pairs of images from the model's predictions. We then used the neural representation similarity to regularize CNNs trained on image classification by penalizing intermediate representations that deviated from neural ones. This preserved performance of baseline models when classifying images under standard benchmarks, while maintaining substantially higher performance compared to baseline or control models when classifying noisy images. Moreover, the models regularized with cortical representations also improved model robustness in terms of adversarial attacks. This demonstrates that regularizing with neural data can be an effective tool to create an inductive bias towards more robust inference.

## 1 Introduction

Convolutional neural network (CNN) models are widely used in computer vision tasks, and can achieve super-human performance on many classification tasks [1, 2]. However, there is a still huge gap between these models and the human visual system in terms of robustness and generalization [3, 4, 5]. In fact, the invariant neural representations and the ability to generalize across complex transformations has been seen as the hallmark of visual intelligence [6, 7, 8, 9]. Understanding why the visual system has superior performance on so many perceptual problems is one of the central questions of neuroscience and machine learning. In particular, CNNs are vulnerable to adversarial attacks and noise distortions [10, 3, 4] while human perception is barely affected by these small image perturbations. This highlights that state-of-the-art CNNs lack human level scene understanding and do not rely on the same causal features as humans for visual perception [4, 5, 11].

Regularization and implicit inductive biases in deep networks can positively affect robustness and generalization by constraining the parameter space and biasing the trained model to use better features. However, these biases are often rather nonspecific and networks often latch onto patterns that do not generalize well outside the distribution of training data. Biological visual systems, however, cope with strongly varying conditions all the time. Based on recently reported overlap between the sensory representations of task-trained CNNs and representations measured in primate brains [12, 13, 14, 15, 16], we thus hypothesized that biasing the representation of artificial networks towards biological stimulus representations might positively affect their robustness.

Here, we show that directly measuring the neural representation in animal visual cortices and biasing CNN models toward a more biological feature space can indeed lead to more robust models. To this end, we recorded the simultaneous responses of thousands of neurons to complex natural scenes in visual cortex of awake mice. In order to bias a CNN towards biological feature representations, we modified its objective function so that convolutional features are encouraged to establish the same structure as neural activities. We found that by regularizing a ResNet [1] towards a biological neural representation, the trained models had higher classification accuracy than baseline when input images were corrupted by random noise or adversarial perturbations. Regularization towards random representations or features from a pretrained VGG model was substantially less helpful.

## 2 Neural representation similarity

We performed several 2-photon scans in primary visual cortex on multiple mice, with repeated scans per animal across different days. During the experiment, the head-fixed mice were able to run on a treadmill while passively viewing natural image each presented for 500ms. In each experiment, we measured responses to 5100 different grayscale images sampled from the ImageNet dataset, 100 of which were repeated 10 times to give 6000 trials in total. Each image was downsampled by a factor of four to $64 \times 36$ pixels. We call the repeated images 'oracle images', because the mean neural responses over these repeated trials serve as a high quality predictor (oracle) for validation trials. The major reason for choosing mice in our study is they allow for genetic tools for large scale recordings ($\sim$8000 units simultaneously). While mice indeed do not have as sophisticated visual systems as primates, vision is still one of their major sensory inputs. Grayscale images were used because mice are not sensitive to the colors relevant to human vision.

We begin by defining the similarity metric for neural responses, which we will then use to regularize a CNN for image classification. In a first step, the raw response $\rho_{ai}$ for each neuron $a$ to stimulus $i$ is scaled by its signal-to-noise ratio

$$w_a = \frac{\sigma_a}{\eta_a} \, , \tag{1}$$

which was estimated from responses to repeated stimuli, namely the oracle images. For a neuron $a$, the signal strength $\sigma_a^2 = \mathrm{Var}_i(\mathbb{E}_t[r_{ait}])$ is the variance over stimuli $i$ of the mean response over repeated trials $t$. The noise strength is the mean over stimuli of the variance over trials, $\eta_a^2 = \mathbb{E}_i[\mathrm{Var}_t(r_{ait})]$. We denote these scaled responses by $r_{ai} = w_a \rho_{ai}$. The scaled population response to stimulus $i$ is the vector $\boldsymbol{r}_i$. Scaling responses based on signal-to-noise ratio accounts for the reliability of each neuron by reducing the influence of noisy neurons. For example, if the responses of a neuron to the same image are highly variable, we will ignore its contribution to the similarity metric by assigning a small weight to it, no matter how differently it responds to different images or how high its responses are in general.

We then shift and normalize these population responses, creating centered unit vectors $\boldsymbol{e}_i = \frac{\boldsymbol{r}_i - \bar{\boldsymbol{r}}}{\|\boldsymbol{r}_i - \bar{\boldsymbol{r}}\|}$ where $\bar{\boldsymbol{r}} = \mathbb{E}_i[\boldsymbol{r}_i]$ is the population response averaged over all stimuli. These unit vectors are then used to construct the similarity matrix, according to

$$S_{ij}^{\mathrm{data}} = \boldsymbol{e}_i \cdot \boldsymbol{e}_j \tag{2}$$

for stimuli $i$ and $j$.

### 2.1 Stability across animals and days

Averaging the responses to the repeated presentations of the oracle images allows us to reduce the influence of neural noise in the representation similarity metric defined in Eq. 2 and examine its

stability across scans (i.e. different selection of neurons). When calculating similarity between oracle images, we can average the results of different trials to reduce noise. For given image $i$ with $T$ repeats, we first treat those trials as if they are different images $i_1, \ldots, i_T$, and calculate similarity against repeated trials of another oracle image $j$, $(j_1, \ldots, j_T)$ in every combination. Oracle similarity is defined as the mean value of all trial similarity values

$$S_{ij}^{\text{oracle}} = \mathbb{E}_{t_i,\, t_j} \left[ S_{i_{t_i} j_{t_j}}^{\text{data}} \right] \,, \tag{3}$$

with $S_{i_t i_t}^{\text{data}} = 1$ excluded when $i = j$.

We found that the neural representation similarity between images is stable across scans and across mice in primary visual cortex (Fig. 1A). When images (columns and rows) are ordered for better visualization, there is a visible structure consistent across scans, revealing the clustering organization of these images. We further index the matrix for scan $h$ as $S_{ij}^{\text{oracle}-h}$, and compare the fluctuation across scans

$$\Delta S_{h,i,j}^{\text{scan}} = S_{ij}^{\text{oracle}-h} - \mathbb{E}_h \left[ S_{ij}^{\text{oracle}-h} \right] \,, \tag{4}$$

and the fluctuation across repeats

$$\Delta S_{h,i,t_1,t_2}^{\text{repeat}} = S_{i_{t_1} i_{t_2}}^{\text{data}-h} - S_{ii}^{\text{oracle}-h} \,. \tag{5}$$

We observer a much narrower distribution for $\Delta S^{\text{scan}}$ than $\Delta S^{\text{repeat}}$ (Fig. 1C), suggesting that the variability due to the selection of neurons (scans) is much lower than the single trial variability to the same image.

## 2.2 Denoising neural responses with a predictive model

Most images in our experiments were only presented once to maximize the diversity of stimuli, so $S^{\text{oracle}}$ is not available for them while $S^{\text{data}}$ was too noisy for our purpose. To exploit the neural responses for non-oracle images, we first train a predictive model to denoise data. The predictive model is consisted of a simple 3-layer CNN with skip connection [17, 18]. It takes images as inputs and predict neural responses by a linear readout at the last layer. In addition, behavioral data such as the pupil position and size, as well as the running speed on the treadmill are also fed to the model to account for the effect of non-visual variables.

The predicted response for neuron $a$ to stimulus $i$ is denoted as $\hat{\rho}_{ai}$, which is trained to predict $\rho_{ai}$ well [18]. The correlation between $\hat{\rho}_a$ and $\rho_a$ is denoted as $v_a$, indicating how well neuron $a$ is predicted. The scaled model response is defined as $\hat{r}_{ai} = w_a v_a \hat{\rho}_{ai}$ with the sigal-to-noise weight $w_a$ from Eq. 1, and the population response is then denoted as $\hat{\boldsymbol{r}}_i$. The similarity matrix for scaled model responses is calculated in the same way as Eq. 2,

$$S_{ij}^{\text{model}} = \hat{\boldsymbol{e}}_i \cdot \hat{\boldsymbol{e}}_j \,. \tag{6}$$

Similarity matrices for the same set of oracle images are shown in Fig. 1B, each from a model trained for the corresponding scan. The similarity for measured neural responses, $S^{\text{oracle}}$, are also present in the model response similarities, but the structure is more prominent for the model responses. A scatter plot of data and model similarities, $S_{ij}^{\text{oracle}}$ versus $S_{ij}^{\text{model}}$ (Fig. 1D), shows a high correlation $r = 0.73$, but the model similarities have a wider range. In the same plot we also showed the correlation between $S^{\text{oracle}}$ and the corresponding trial similarity values $S^{\text{data}}$ from which they are estimated, and found $S^{\text{model}}$ to be much less noisy than $S^{\text{data}}$.

The use of model neuron responses as a proxy for the real neurons has three major benefits. First, the outputs are deterministic, eliminating the random noise component. Second, the predictive model was heavily regularized during training, so these deterministic responses are more likely to reflect reliable visual features. Third, the model's shifter and modulator circuit [17] accounted for the irrelevant nonvisual eye and body movements, and could thereby extract more of the purely visual-driven responses.

With the help of a predictive model, we can obtain cleaner responses for the 5000 non-oracle images even though they are only measured once. We used the similarity matrices averaged over 8 scans as the regularization target. Two examples of the model neural similarity for the 100 oracle images are shown in Fig. 2. It is worth clarifying that we don't use this 100×100 matrix in our main result though, but only the 5000×5000 matrix from non-oracle trials. Oracle trials are used for evaluating predictive models, assigning neuron-specific weights and demonstrations (Fig. 1 and 2) only.

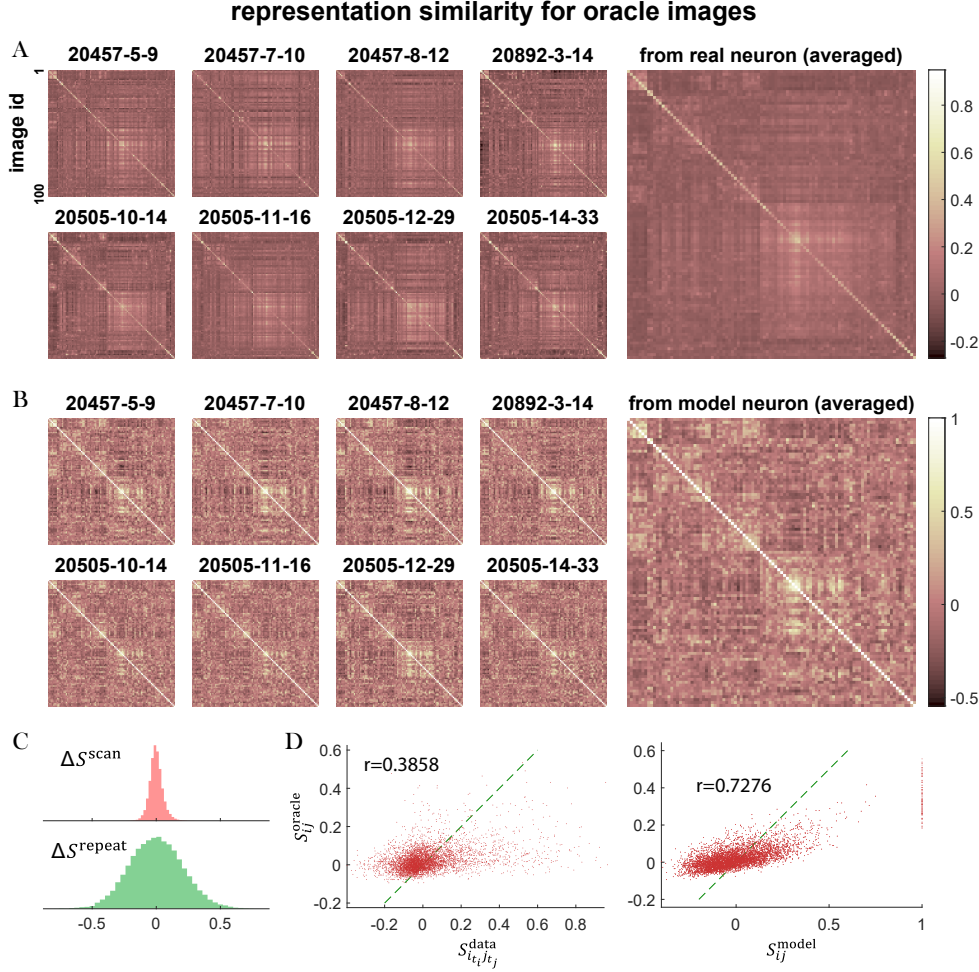

Figure 1: Representation similarity in neural data and predictive models. Image order is preserved across all matrices. (A) Similarity matrices $S^{\text{oracle}}$ (Eq. 3) from real neural responses to 100 oracle images. The structure of similarity matrices is stable across scans on different animals and days. (B) Similarity matrices $S^{\text{model}}$ (Eq. 6) from predictive models. The backbone of representation similarity is preserved and enhanced. (C) Variability over scans is smaller than that over repeats. (D) Similarity values estimated from single trials are noisy ($S^{\text{data}}$ vs. $S^{\text{oracle}}$, $r = 0.39$), but similarity calculated from predictive models correlate with those from neural data well ($S^{\text{model}}$ vs. $S^{\text{oracle}}$, $r = 0.73$). In addition, $S^{\text{model}}$ spans a wider range than $S^{\text{oracle}}$, makes it a better training target in practice.

## 3    Neural regularization by joint training

To regularize a standard machine learning model with the representation similarity matrix obtained from neural data [19], we jointly train the model with a similarity loss in addition to its original task-defined loss (Fig. 3, also see [20] and [21] for related approaches based on fMRI or other deep neuronal networks, respectively). The full loss function contains two terms, defined as

$$L = L_{\text{task}} + \alpha L_{\text{similarity}} . \tag{7}$$

The first term is a conventional loss used to define the performance on the task, such as classification or 1-shot learning. In this section, we implement grayscale CIFAR10 classification, hence we use a cross-entropy loss. The second term is the penalty that favors brain-like representations, with a coefficient $\alpha$ determining regularization strength.

For any pair of images that were shown to the mice, we already have their representational similarity from models predicting neural data (Eq. 6). Since we are now comparing similarity for two models, a

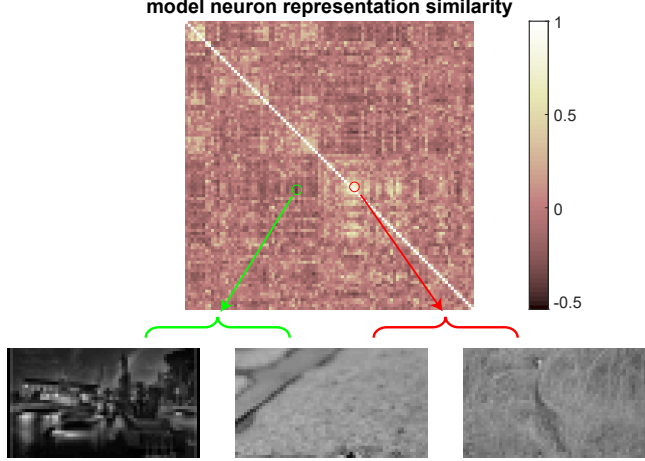

Figure 2: Examples of similar and dissimilar image pairs. From the similarity matrix of model neural responses, examples of low similarity value (left pair) and high similarity value (right pair) are shown.

neural predictive model and a task model based on a convolutional neural network, we denote the former by $S_{ij}^{\mathrm{neural}}$ and the latter by $S_{ij}^{\mathrm{CNN}}$. We want $S^{\mathrm{CNN}}$ to approximate $S^{\mathrm{neural}}$ well.

We define the similarity loss for image $i$ and image $j$ as

$$L_{\mathrm{similarity}} = \left[\mathrm{arctanh}\left(S_{ij}^{\mathrm{CNN}}\right) - \mathrm{arctanh}\left(S_{ij}^{\mathrm{neural}}\right)\right]^2 . \tag{8}$$

The $\mathrm{arctanh}$ is used to remap the similarities from the interval $[-1, 1]$ to $(-\infty, \infty)$. It is analogous to the Fisher transform which uses the same $\mathrm{arctanh}$ function to compute confidence intervals for correlation coefficients, by reparameterizing the correlations to follow nearly normal distributions. When similarity values are not too close to $-1$ or $1$, the loss is close to the sample based centered kernel alignment (CKA) index [22, 23, 24].

Intuitively, $S^{\mathrm{CNN}}$ is the cosine similarity of convolutional features that image $i$ and $j$ activate. Though V1 responses are thought to encode low-level features, there's no principled way to determine *a priori* which single model layer corresponds to V1. Thus we flexibly combine feature similarities from a selection of layers instead of assigning to a specific one. Specifically, we calculate similarity for $K$ uniformly located convolutional layers, and average the results through a trainable weight. The weights are the outputs of a softmax function, therefore guaranteed to be positive and sum to one. Mathematically speaking, for each of the $K$ layers we compute the cosine similarity values as

$$S_{ij}^{\mathrm{CNN}-k} = \frac{\left(\boldsymbol{f}_i^{(k)} - \bar{\boldsymbol{f}}^{(k)}\right) \cdot \left(\boldsymbol{f}_j^{(k)} - \bar{\boldsymbol{f}}^{(k)}\right)}{\left\|\boldsymbol{f}_i^{(k)} - \bar{\boldsymbol{f}}^{(k)}\right\| \left\|\boldsymbol{f}_j^{(k)} - \bar{\boldsymbol{f}}^{(k)}\right\|}, \tag{9}$$

where $\boldsymbol{f}_i^{(k)}$ is the concatenated convolutional feature vector for image $i$ at layer $k$, and $\bar{\boldsymbol{f}}^{(k)} = \mathbb{E}_i\left[\boldsymbol{f}_i^{(k)}\right]$ is its mean over images. The final model similarity is a combination from all selected layers

$$S_{ij}^{\mathrm{CNN}} = \sum_k \gamma_k S_{ij}^{\mathrm{CNN}-k}, \tag{10}$$

where $\gamma_k$ is a trainable probability with $\sum_k \gamma_k = 1$, $\gamma_k \geq 0$. This means that the objective function can choose which layer to match the similarity, but it needs to match at least one in total as enforced by the softmax that determines $\gamma_k$. In principle all convolutional layers can be included, but we only used 5 in our simulations (layer 1, 5, 9, 13, 17 in ResNet18). The preliminary analysis shows after training one layer will dominate in Eq. 10, and it is typically layer 5, the last layer of the first ResBlock group. More details are included in the supplementary materials.

In each step of training, we first process a batch of CIFAR images to calculate classification loss $L_{\mathrm{classification}}$, and subsequently process a batch of image pairs sampled from the stimuli we used

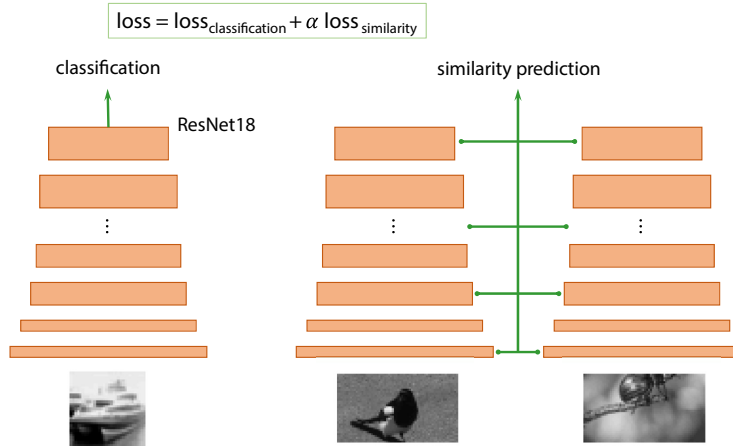

$$\text{loss} = \text{loss}_{\text{classification}} + \alpha \, \text{loss}_{\text{similarity}}$$

Figure 3: Joint training schematic. We trained a ResNet18 model to both classify CIFAR10 images and predict neural similarity of ImageNet images used in our scan. The network takes either one image or a pair of images as inputs, with a same convolutional core. If the input is one image with the right size, the model outputs class prediction with an additional fully connected layer. If the input is a pair of images, the model first calculate the convolutional features for both, and calculate the similarity for a few selected layers (Eq. 10). Similarity predictions from different layers are summed up by a trainable normalized weight to produce a final prediction, which is trained to match neural similarity (Eq. 6). Two losses are summed with a coefficient $\alpha$ as the regularization strength.

in experiments, calculating the similarity loss $L_{\text{similarity}}$ with respect to the pre-computed $S^{\text{neural}}$ matrix. The gradient of the full loss can affect the CNN kernel weights through both loss terms.

## 4 Results

### 4.1 Robustness against random noise

The similarity loss plays the role of a regularizer, and it biases the original CNN towards a more brain-like representation. We observed that the CNN model becomes more robust to random noise when neural regularization is used. Compared to a ResNet18 [1] trained without any regularization ('None' in Fig. 4A), the same architecture equipped with the neural regularizer ('Neural (model)' in Fig. 4) had substantially better performance on noisy input images ($\sim$50% v.s. $\sim$20% at the highest noise level). In other words, models whose features are more neural are less vulnerable to random noise in inputs. To strengthen this conclusion, we also regularized the model with shuffled $S^{\text{neural}}$ matrix ('Shuffle' in Fig. 4) or the feature similarity matrix of the `conv3-1` layer in a VGG19 model pretrained on ImageNet ('VGG' in Fig. 4). This VGG layer has been reported to be most similar to animal V1 [16]. Both regularizers improve the model robustness to some degree but neither as much as using the neural regularizer.

Finally, we also regularized the model with a similarity matrix from the actual data directly ('Neural (data)' in Fig. 4), using $S^{\text{data}}$ (Eq. 2) instead of $S^{\text{model}}$ (Eq. 6). We did not observe the same boost in robustness. We think that this is caused by the high variability of the neural responses, highlighting the need for a well trained predictive model. In addition, if we see the matrix in 'Shuffle' control as the feature similarity of a poorly trained predictive model, the difference between 'Neural (model)' and 'Shuffle' again shows the importance of having a well trained one. Only with a strong predictive system identification model as a denoiser were we able to reveal the underlying representational structure hidden in the noisy neural data.

We observed the same results when training ResNet34 models on grayscale CIFAR100 datasets (Fig. 4B). In addition, we also tested how different regularization strength will affect the model performance, and observed a continuous increase of model robustness when we tuned up the regularization. More details are included in the supplementary materials.

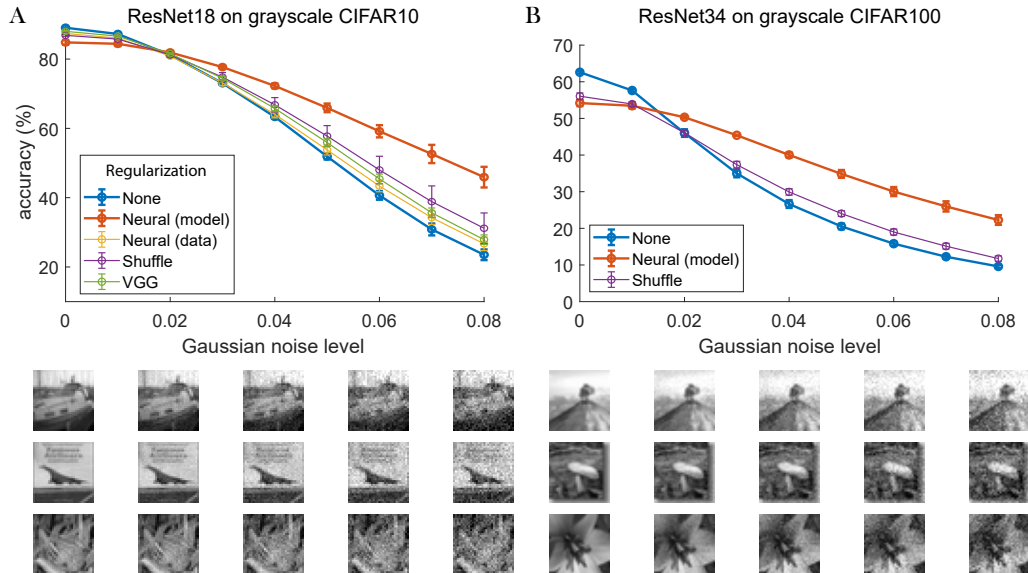

Figure 4: Performance robustness to Gaussian noise. (A) We tested CIFAR10 classification performance under different levels of Gaussian noises on input images (examples below the plot) for our jointly trained ResNet model, and compared with models with no regularization and some other regularization. Compared to the vanilla network with no regularization ('None'), all regularized model have higher classification accuracy when discernible noise is added. In particular, the model regularized with model neural similarity outperforms others on noisy images, only with a small sacrifice on clean image performance. The error bars here are standard error of mean (SEM), with 5 random seeds used for each regularizer. The reduced improvement from 'Neural (data)' emphasizes the need for a good predictive model for denoising, so that the actual neural representation structure can be exploited. (B) Results for ResNet34 on grayscale CIFAR100 dataset are shown for 'None', 'Neural (model)' and 'Shuffle'.

All models are trained by stochastic gradient descent for 40 epochs with batch size 64. Learning rate starts at 0.1 and decays by 0.3 every 4 epochs, but resets to 0.1 after the 20th epoch. Mean classification accuracy for CIFAR10/100 test set over 5 random seeds is reported in Fig. 4. In our current setting, the same number of images are passed in the classification pathway and neural pathway, hence the time cost approximately doubles comparing to normal training. It takes about 4.5 hours on a single TITAN RTX GPU to train one model. We used PyTorch [25] for model training.

## 4.2 Robustness against adversarial attack

We are also interested in whether neural regularization provides robustness to adversarial attacks. Since adversarial examples and their innocent counterparts elicit the same percept by definition, it is highly possible that their measured neural representations are also close to each other. Hence a model with neural representation will be more invariant to adversarial noise. We evaluated model robustness following a recently published guideline [26] and using the well-tested attack implementations provided by Foolbox [27].

Our evaluation metric follows [28]. In a nutshell, we strive to find adversarial perturbations (*i.e.* perturbations that flip the label to any but the ground-truth class) with the minimum norm (either $L_2$ or $L_\infty$) for each of 1000 test samples. We then compute the median perturbation distance across all samples as the final robustness score (higher is better).

Besides the current state-of-the-art attacks on $L_2$ [26] and $L_\infty$ [29], we also deployed a recently developed gradient-based version of the decision-based boundary attack [30], which surpasses [26] in terms of query efficiency and the size of the minimal adversarial perturbations. In short, [30] starts from a natural input sample that is classified as different from the original image (for which we aim to generate an adversarial example). The algorithm then performs a line search between the two images

to find the decision boundary of the model. The gradients w.r.t. the difference between the two top-most logits allow us to estimate the local geometry of the decision boundary. Using this geometry we can compute the optimal adversarial perturbation that (a) takes us exactly to the boundary (in case we are slightly shifted away from it), (b) stays within the valid pixel bounds, (c) minimizes the distances to the original image, and (d) is not too far from the current perturbation (to make sure we stay within the region for which the linear approximation of the boundary is valid). Therefore, our gradient-based version of the decision boundary attack provides a most stringent test for adversarial robustness of our neural network machine learning models regularized with neural data.

To ensure that we evaluate and compare all models fairly, we perform an extensive hyperparameter search and always select the optimal combination. Since our gradient-based boundary attack proved more effective than [26] on all models tested here, we only deployed the gradient-based boundary attack for $L_2$, and used Projected Gradient Descent (PGD) [29] for $L_\infty$ in our final evaluation. For our gradient-based boundary attack, we tested step sizes of $\{0.0003, 0.001, 0.003, 0.01, 0.03, 0.1, 0.03\}$, and for PGD we tested step sizes of $\{10^{-6}, 10^{-5}, 10^{-4}, 10^{-3}, 10^{-2}, 10^{-1}, 1\}$ with iterations of $\{10, 30, 50, 100, 200\}$.

Fig. 5 shows that regularizing models with neural representational similarity improves model robustness against adversarial attacks. The model with the smallest adversarial perturbations (most fragility) is the vanilla model trained without any regularization (median perturbation of $0.0025$ ($L_\infty$) and $0.09$ ($L_2$)). Regularizing with random similarity matrix (median perturbation of $0.003$ ($L_\infty$) and $0.11$ ($L_2$)) or similarity of VGG features (median perturbation of $0.0028$ ($L_\infty$) and $0.11$ ($L_2$)) increases robustness. The strongest increase in robustness, in both metrics, is provided by the regularization with the brain's representations learned from neural data (median perturbation of $0.0034$ ($L_\infty$) and $0.13$ ($L_2$)).

We additionally did a more thorough experiment with a few more type of attacks, and looked at $L_0$ and $L_1$ metrics. More details are included in the supplementary materials.

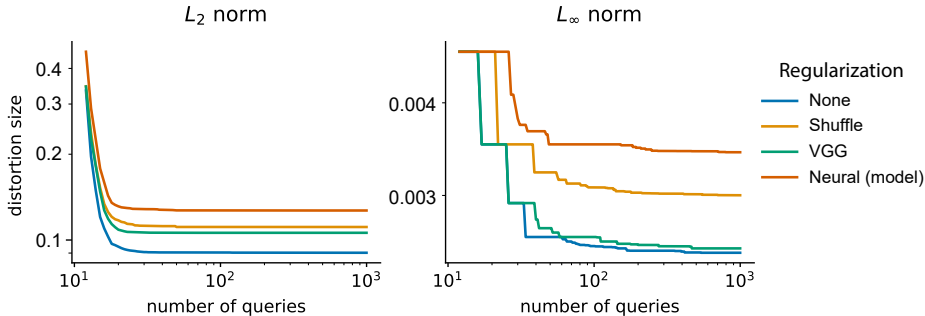

Figure 5: Adversarial robustness of classifier networks according to the $L_2$ and $L_\infty$ norms. With more optimization queries for the attack, the minimal perturbation shrinks. Regularization improves adversarial robustness, with neural regularization providing the best defense throughout the attacker's optimization. 'Neural (data)' regularizer is not tested.

We observed increased robustness across several metrics, which is quite remarkable given that current defense methods, in particular adversarial training, tend to overfit strongly on the metrics on which they are trained on [28] and are often less robust on other $L_p$ metrics than undefended models.

## 5    Conclusion and discussion

Neuroscience has often provided inspiration to machine learning, but it lacks methods to directly translate neurophysiological recordings into an improvement of artificial neural networks. Here, we have shown that regularization with neural data proves to be a promising tool to create an inductive bias towards more robust inference. In particular, the mouse brain has evolved an image representation that is capable of performing difficult machine learning tasks, but is more robust than conventional models. Specifically, we demonstrated that when these measured representations are incorporated into a general machine learning model by matching representational similarity, they enable more robust machine learning algorithms (robustness to random noise and adversarial attacks).

Critically, our predictive computational model provided a better assessment of representation than raw neural data, because it disentangles non-visual features from visual ones, and transforms the isolated visual features into a reliable, de-noised version of neural responses. This model-based representation proved useful as a regularization target for machine learning models. We conjecture that by regularizing machine learning models further to match representational similarities with higher order visual areas beyond V1 will further enhance the robustness and generalization performance outside the training set. These brain-like representations may help machine learning algorithms ultimately reach human-like performance.

There are at least two ways to regularize CNN models to favor neural representations. One is to learn the similarity for any pair of images, like our approach here. The other is to jointly train a linear readout from intermediate layers of CNN to predict neural responses directly. However, we argue that the former is a tighter constraint since a wide range of affine transformations in the CNN could be compensated by the linear readout, producing identical predictions for the neural responses while substantially altering the underlying representational similarity in the CNN. For this reason, we chose to regularize our machine learning models to match the representational similarity.

Though the improvement of adversarial robustness by neural regularization is substantial and significant, unsurprisingly, the current state-of-the-art in terms of robustness on $L_\infty$ [29] remains substantially more robust than our neurally regularized but otherwise undefended model (0.029 vs 0.0034). That said, [29] employs an expensive adversarial training procedure that—in contrast to our method—specifically aims to optimize robustness against $L_\infty$ perturbations. As a side effect, [29] performs significantly worse on metrics it has not been trained on, such as $L_2$ or $L_0$ [28] while our method does not overfit on one specific metric. Combining the neural regularization with adversarial training procedure [29] could potentially lead to even stronger defenses.

The neural regularization is not designed to improve model robustness, but rather to bias any model to have neural features. We expect to see other benefits with such inductive bias, such as improved generalization in domain transfer, lower sample complexity in few-shot learning, and so on. While more systematic analysis is continuing, preliminary results indeed have shown improvement by neural regularization in those aspects as well.

To bias CNN features towards a more brain-like representation, we matched the pairwise cosine similarity for a given set of inputs in this study. But this is just one approach of manifold matching in a more general sense [31]. We will explore other metrics or higher-order dependencies in the future.

While our results indeed show the benefit of adopting more brain-like representation in visual processing, it is however unclear which aspects of neural representation make it work. We think that it is *the* most important question and we need to understand the principle behind it. There are two approaches that we are currently working on. The first is to directly compare the regularized models and the vanilla ones by investigating the features they use. We will look into the tuning property of model units by finding the input patterns that maximally excites these units, and examine how neural regularization makes a difference. The second is to identify which neurons are useful for a more robust representation. We can either find the subset of neurons that are most important in the similarity metric and look for their common properties. Or we can propose some criterion to select a particular set of neurons and check whether using those neurons alone can obtain the same robustness gain. If we manage to understand why neural regularization works, we'll be able to design or train machine learning models just with the underlying principles, without actually performing large-scale neural recordings.

A docker image containing all codes and trained models is prepared (zheli18/neural-reg:neurips19), with a jupyter lab as entrypoint.

## Acknowledgements

This work is supported by the Intelligence Advanced Research Projects Activity (IARPA) via Department of Interior/Interior Business Center (DoI/IBC) contract number D16PC00003. The U.S. Government is authorized to reproduce and distribute reprints for Governmental purposes notwithstanding any copyright annotation thereon. Disclaimer: The views and conclusions contained herein are those of the authors and should not be interpreted as necessarily representing the official policies or endorsements, either expressed or implied, of IARPA, DoI/IBC, or the U.S. Government. FS

is supported by the Institutional Strategy of the University of Tübingen (Deutsche Forschungsge-meinschaft, ZUK 63) and the Carl-Zeiss-Stiftung. FS and MB acknowledges the support from the German Federal Ministry of Education and Research (BMBF) through the Tübingen AI Center (FKZ: 01IS18039A) and the DFG Cluster of Excellence "Machine Learning – New Perspectives for Science", EXC 2064/1, project number 390727645.

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
