[Supplementary Material]

# Learning From Brains How to Regularize Machines (Supplementary Material)

**Zhe Li[1-2,*], Wieland Brendel[4-5], Edgar Y. Walker[1-2,7], Erick Cobos[1-2],**
**Taliah Muhammad[1-2], Jacob Reimer[1-2], Matthias Bethge[4-6],**
**Fabian H. Sinz[2,5,7], Xaq Pitkow[1-3], Andreas S. Tolias[1-3]**

[1] Department of Neuroscience, Baylor College of Medicine
[2] Center for Neuroscience and Artificial Intelligence, Baylor College of Medicine
[3] Department of Electrical and Computer Engineering, Rice University
[4] Centre for Integrative Neuroscience, University of Tübingen
[5] Bernstein Center for Computational Neuroscience, University of Tübingen
[6] Institute for Theoretical Physics, University of Tübingen
[7] Institute Bioinformatics and Medical Informatics, University of Tübingen

[*]zhel@bcm.edu

## 1 Robustness dependence on regularization strength

The training objective is a combination of task loss and similarity loss, with a relative weight $\alpha$ in $L = L_{\text{task}} + \alpha L_{\text{similarity}}$. We tested a range of $\alpha$ values, and observed a continuous change in model performance (Fig. 1). Here similarity matrix is estimated using just one scan, while results in the main text are using averaged similarity matrix from eight scans. $\alpha = 20$ was used in the main text, qualitatively same as the $\alpha = 16$ shown here. For each $\alpha$, 2 or 3 random seeds were used.

Figure 1: Robustness to Gaussian noise at different regularization strengths. $\alpha = 0$ is the 'None' condition in main text, which is mostly occluded by $\alpha = 2$ here. As neural regularization is more strongly applied, the model performance on noisy inputs becomes higher.

## 2 Combination weights for CNN model similarity

We used a trainable weight $\gamma_k$ (Eq. 10 in main text) to combine feature similarity of different convolutional layers to the final similarity of the full model. We design $\gamma_k$s to be the outputs of a

softmax function, and have the same initial values. In our simulations, $K = 5$ layers are selected, hence $\gamma_k = 0.2$ for $k = 1, 5, 9, 13, 17$ in the beginning of training.

We observed that after joint training, $\gamma_k$ usually collapse to only one layer. Namely $\gamma_k \approx 1$ for one layer, and close to 0 for the others. We think this is a direct result from the competitive nature of our weight design. As long as one layer is selected to resemble the neural feature space, the joint training algorithm will keep pushing it towards the target. The identity of the selected layer, which is usually the easiest one to adjust to neural feature space, is not deterministic. We investigated final weights for models in Fig. 1, and the averaged weights are listed in Tab. 1.

Table 1: Averaged weights for all candidate layers.

|  | $\gamma_1$ | $\gamma_5$ | $\gamma_9$ | $\gamma_{13}$ | $\gamma_{17}$ |
|---|---|---|---|---|---|
| $\alpha = 0$ | 0.2 | 0.2 | 0.2 | 0.2 | 0.2 |
| $\alpha = 2$ | 0 | 1 | 0 | 0 | 0 |
| $\alpha = 4$ | 0 | 1 | 0 | 0 | 0 |
| $\alpha = 8$ | 0 | 0.33 | 0.67 | 0 | 0 |
| $\alpha = 16$ | 0 | 0.67 | 0.33 | 0 | 0 |
| $\alpha = 32$ | 0.5 | 0.5 | 0 | 0 | 0 |

For example, $\gamma_5 = 0.67$ for $\alpha = 16$ actually corresponds to that 2 out of 3 random seeds result in a trained model with $\gamma_5 = 1$. Though there exists stochasticity in the choice of layers, the possible ones are usually nearby in terms of their locations in the deep network. Admittedly, more simulations are needed to be conclusive.

## 3 More extensive tests on adversarial robustness

We performed a much more thorough tests on our trained models with two more metrics and six more attacks after the submission. The models being tested here ('None', 'Shuffle' and 'Neural') are also newer version since we improved the neural predictive model since then. In short, more reliably measured neurons are weighted even more now, which in theory makes the neural similarity matrix less noisy.

The evaluation of the models follows the evaluation scheme of [1]. We tested all models on four different $L_p$ metrics ($L_0, L_1, L_2$ and $L_\infty$) with different state-of-the-art attacks (see below). Every model/attack combination was evaluated on 1000 samples from the CIFAR-10 validation set and we used the same subset for all models. Then, on each sample and on each model/attack combination each attack was run five times for each hyperparameter setting we tested in an untargeted attack scenario. For each attack we tested a range of hyperparameters to ensure optimal performance. We used attacks as implemented in Foolbox [2]. To gather the final distortion sizes shown in Fig. 2 we determined the smallest $L_p$ distance for each sample and for each model/attack combination across all tested hyperparameters and repetitions. We hope that this scheme approaches as closely as possible the true minimal adversarial distance. We then average this minimal adversarial distance over all 1000 samples to determine model robustness.

Across $L_1, L_2$ and $L_\infty$ we observe a market increase in robustness compared to baseline and control networks. This increase is unlikely to be caused by gradient-masking given that adversarial attacks work equally well on all models on the $L_0$ norm. At the same time, $L_0$ is also a special metric in the sense that it introduces strong deviations between original and adversarial image which are also the most noticeable for humans.

The attacks that we applied to the models are as follows:

- *Projected Gradient Descent (PGD) [3].* Iterative gradient attack that optimizes $L_\infty$ by minimizing a cross-entropy loss under a fixed $L_\infty$ norm constraint enforced in each step.
- *Projected Gradient Descent with Adam (AdamPGD) [4].* Same as PGD with but Adam Optimiser for update steps.
- *C&W [5].* $L_2$ iterative gradient attack that relies on the Adam optimizer, a tanh-nonlinearity to respect pixel-constraints and a loss function that weighs a classification loss with the distance metric to be minimized.

Figure 2: Adversarial robustness for ResNet18 models on grayscale CIFAR10 dataset. Models with no regularization ('None'), regularized to shuffled similarity matrix ('Shuffle') and to neural similarity matrix ('Neural') are tested for four metrics. Neural regularization increased model robustness to adversarial perturbations for $L_0$, $L_1$ and $L_\infty$.

- *Decoupling Direction and Norm Attack (DDN) [6].* $L_2$ iterative gradient attack pitched as a query-efficient alternative to the C&W attack that requires less hyperparameter tuning.
- *Saliency-Map Attack (JSMA) [7].* $L_0/L_1$ attack that iterates over saliency maps to discover pixels with the highest potential to change the decision of the classifier.
- *Sparse-Fool [8].* A sparse version of DeepFool, which uses a local linear approximation of the geometry of the decision boundary to estimate the optimal step towards the boundary.
- *Brendel&Bethge [1].* Novel family of $L_0/L_1/L_2/L_\infty$ attacks that follow the boundary between the adversarial and non-adversarial region which has been demonstrated to be state-of-the-art on all tested $L_p$ norms.