[Reviews · NeurIPS 2019]

Reviewer 1



CNNs, like visual cortex, build a representation of the visual world that is useful to the “viewer”. We have known for a while that CNNs trained on object recognition tasks capture some (but not all) aspects of the representation computed by primate visual cortex. Here the authors propose to bridge the gap by explicitly encouraging a CNN to build a representation that is “similar” to the one computed by the visual cortex of mice. This is a neat idea and certainly a novel one. The paper is clearly written, which I appreciated. The research question being tackled is clearly explained, the experiments are properly designed (I really like the randomized matrix control). The research being presented is properly placed in the context of existing literature and all the moving parts are well summarized, which made the paper very easy to follow. I wish the authors would consider citing at least one piece of work by Poggio and collaborators. Poggio spent the first half of this decade characterizing the robustness of neural and artificial representations, so I think that line of inquiry is relevant. Maybe either Tacchetti, Isik, Poggio Annual Reviews of Vision Science 2018, which is a good review of that work, or the book Poggio, Anselmi 2016 with MIT Press. I genuinely liked this paper, and I think it presents a very interesting idea that I am sure will inspire further inquiry. There are a few things I wish the authors had included, and hopefully might serve as a suggestion for a revised camera ready version, and which might improve the significance of this work. 1) While this paper certainly presents a “cool” idea, it is unclear what we, as either neuroscientists trying to understand the brain, or computer scientist trying to replicate human visual intelligence, are to make of this result. Is there a way to dive deeper and understand which nuances the unregularized CNN representation was missing? What is the “computational goal” encoded in the mice similarity matrix that was not present in the baseline CNN representation? How does this result help us build better CNNs, or what did we learn about the mouse visual system that we did not know before? 2) One possible way to start getting at this, would be investigating what happens if one trains the CNN exclusively with the regularization term (no task). Is that representation worse? Where does it fail? 3) In a similar spirit, random noise and adversarial perturbations are solid choices for a sanity check, but do not really reveal where the two representations differ and why. I wish the authors had designed more semantically relevant “attacks”. Maybe it’s 3-D rotations, maybe it’s illumination, what is it that CNN are missing. 4) Finally, a minor suggestion: it is my understanding that mice are pretty much blind, and either way do not “use” their sense of vision all that much. Why did you choose mice? Could you put a sentence or two in the text to explain your choice? Thank you for sharing these cool ideas and results! I hope these suggestions help. All the best!

Reviewer 2



This paper introduces a neural regularization method for CNN based image classification architectures. The authors hypothesize that biasing the representation of artificial networks towards biological stimulus representations might positively affect their robustness. They present natural images (CIFAR10) to mice and measured the responses of thousands of neurons from cortical visual areas. A prediction model is trained on the collected data (100 oracle images), in order to estimate the neuron responses to a much larger image set (5000 images) and perform denoising. The neural representation similarity is then used to regularize CNN by penalizing intermediate representations that are deviated from neural ones. Experimental results on the CIFAR10 data show that the proposed regularization method can achieve better performance on classifying noisy images, in comparison with several baselines or control models. (1) The idea of using the similarity of neurophysiological data to regularize the representation of artificial neural networks is interesting and novel. The authors have made good efforts towards bridging the fields of neuroscience and machine learning through such regularization. (2) Experimental results on the CIFAR10 data demonstrate the effectiveness of the proposed method and provide some validation of the hypothesis. (3) In the proposed joint training method (Section 3), a selection of layers from the bottom to the top of the architecture is chosen. The final similarity of representations from CNN is a weighted average of the similarity calculated from each layer. The output from different layers can be dramatically different, which lead to quite different final similarity values. The authors need to provide guidelines on how to choose such a selection of layers. (4) In Section 2.2, a predictive model is used to "denoise neural responses". What is the prediction accuracy (or correlation) in the proposed experiment setup (i.e., use the neural response to 100 oracle images to predict the response to the non-oracle 5000 images)? The scaled model response is defined as $\hat{r}_{ai}=w_{a}v_{a}\hat{rho}_{ai}$. It is not clear which correlation measure is used to compute $v$. Generally speaking, correlation coefficient measures the statistical relationship between two variables. The definition of the scaled model response is not mathematically correct and meaningful. I am a bit concerned about how the performance of the predictive model can affect the final results. The prediction accuracy of a model built on 100 images is likely not high when applied to 5000 images. (5) The proposed method is only applied to one dataset CIFAR10 and one architecture ResNet18. The authors may want to report the performance on other datasets and architectures as well. (6) The authors claim that "we denoised the notoriously variable neural activity using strong predictive models trained on this large corpus of responses from the mouse visual system, and calculated the representational similarity for millions of pairs of images from the model’s predictions". From the current experiment setup, the prediction model is only applied to 5000 non-oracle images, not on million scale. (7) How can the proposed neural regularization method be used in practice? It is apparently not a trivial effort to collect actual neural responses, build prediction model, and then perform joint training. It would be great if the authors can provide some discussions along this line. (8) Minor issue: there are also some typos and grammar issues in the paper. Post-Rebuttal: I appreciate the authors' response, which helped clarify some important technical details, especially on the use of 100 oracle images, the training of predictive models on 5000 non-oracle images, as well as the scale of training data. Authors also added new expriments on other datasets and architectures. I hope these clarifications and discussions can be included in the final version of the paper.

Reviewer 3



It is a very meaningful topic to use the real biological stimulus to help neural network training. This paper proposes an interesting idea on regularizing NN by preserving the feature distance measured by mice's brain. If it is the first paper on doing this, I really think it would be an interesting paper to appear on NeurIPS. However, the paper does have many drawbacks. First, this paper does not have a related work section and does not provide enough survey on the prior work on this domain. It leads to another issue of the paper, not stating the contribution clearly. For example, does this paper is the first to use the mice brain's stimulus to regularize NN? For the technique part, some design is not well motivated. 1. Why use grayscale images instead of RBG images? Is it because mice not recognize color? 2. Why need 'oracle images'? This part is very confusing. Why some images have quality predictor? How do you pick those 'oracle images'. Please explain. 3. The design of the similarity of convolutional features is also tricky. Why the linear coefficient (equation 10) to combine the similarities in each layer is trainable? What if I just take the average of the layer similarities as the convolutional feature similarity? Minor comments: Although it is an interesting method, it seems not very scalable since I have to show images to animals to get regularization data. Also how the amount of the regularization data affects the regularization result. What if I use more data, will the result be better?

[Author Response · NeurIPS 2019]

We thank all three reviewers for their constructive and valuable feedback. They found our paper to be a "very interesting
idea" (R1), to be a "good effort towards bridging the fields of neuroscience and machine learning" (R3), and to cover a
"very meaningful topic" (R4). Their main concerns are to clarify experimental and numeric details and choices, as well
as the implications for neuroscience and ML. We think that our comments below and several new analysis which we
will include in the final version will clarify all open questions, and address most requested improvements.

**R1: Principles behind the success of neurally regularized ML?** We agree that this is *the* central question, and this
is ongoing work. We do not have a final answer yet, but we will discuss some hypotheses in the final version.

**R1: Training with the regularization alone.** We performed related experiments in which we increased the relative
importance of the similarity regularization over the classification loss ($\alpha$ in Eq. 7). Stronger regularization yields greater
robustness, at the cost of worse classification on clean images. We will include these results in supplementary material.

**R1: Consider testing semantically relevant perturbations.** We fully agree. We did test models with/without neural
regularization on a great variety of distortions (Fig. 1b). In all cases we find a better robustness of the regularized
network, although to varying degrees depending on the distortion.

(a) ResNet34 on grayscale CIFAR100 (b) Generalization performance on perturbed 16-class ImageNet

Figure 1: (a) Robustness test of ResNet34 models on CI-FAR100. (b) Generalization result to naturalistic perturbations on 16-class ImageNet dataset.

**R1: Why mice?** While mice indeed do not have the sophisticated visual systems of primates, vision is still one of their
major sensory inputs. Experimentally, mice allow for genetic tools for large scale recordings ($\sim$8000 units).

**R3, R4: Selection of model layers to be regularized.** Though V1 responses are thought to encode low-level features,
there is no principled way to determine *a priori* which single model layer corresponds to V1. Thus we combine
similarities from uniformly located layers (1, 5, 9, 13, 17 in ResNet18) flexibly without assigning to a specific one. Our
model learns which are the optimal layers to regularize and find typically one lower layer.

**R3: How can a predictive model trained on 100 images generalize to 5000?** There is some misunderstanding here:
the predictive model is trained on 5000 non-oracle images with the raw responses $\rho_{ai}$ as targets. The 100 oracle images
are used for evaluating the predictive model only (*e.g.* Fig. 1D in main submission).

**R3: Correlation measure.** The predictive quality weights $v_a$ (line 96-97) are the correlation coefficients estimated
over the 100 oracle (test) images. They bias the regularization toward similarities from better predicted neurons.

**R3: Influence of prediction performance.** The used models are highly predictive and state of the art ([13, 14],
Ecker2019 ICLR), and we find their denoising effect to be crucial to reliably transfer robustness from experimental data.

**R3: Other datasets and architectures?** We reproduced our results on ResNet34 on CIFAR100 and a subset of
ImageNet [3] (Fig. 1). We will include the results in the final version.

**R3: You used 5000 images, not millions.** We are referring to all pairs of images and the corresponding similarity
targets, which are around $\frac{1}{2}5000^2 \approx 12.5$M training samples.

**R3: Use this technique without new experiments?** Our predictive model generalizes across stimuli [13], letting us
compute similarities without experiments. Future work on understanding why neurons improve robustness will also
provide design principles without new experiments.

**R4, R1: Previous work.** To the best of our knowledge, we are the first to demonstrate that similarities derived from
cellular level neural activities improves robustness in machine learning of any sort. Other work used similarity based
regularization or fMRI activity, and we cite them in line 120 (Refs. [16, 17]). We will include a reference to Poggio.

**R4: Why gray scale images?** Mice are not sensitive to the colors relevant to human vision.

**R4: Oracle images?** Neural responses to the same stimulus are not exactly the same due to noisy responses. To
estimate the reliability of each neuron we use randomly picked (oracle) images which we presented repeatedly during
the experiment. When regularizing, each neuron contributed to the similarity measures according to its reliability.

**R4: How many neurons?** We are currently investigating this. However, we find the similarity matrices to be robust
w.r.t. to the selection of neurons (Fig. 1 in main submission).

[Meta-Review · NeurIPS 2019]

The authors demonstrate how to train a CNN to be more robust to random noise perturbations and adversarial attacks by regularizing the network to match the similarity matrix accrued from neural recordings from mice. Building robust CNNs is an extremely important application and leveraging ideas from neuroscience to make this happen is a clever and exciting intersection of the fields. Reviewers commented that the paper was clearly written; the experiments are properly designed and controlled; and the research questions is crisp. There was some minor issues that need to be addressed as the reviewers request including an improved related work section. Assuming all of the issues mentioned by the reviewers are addressed, this paper will be accepted into this conference.